# Real-Time ISR-YOLOv4 Based Small Object Detection for Safe Shop Floor in Smart Factories

Byungjin Ku [1], Kangsan Kim [2] and Jongpil Jeong [1,*]

1. Department of Smart Factory Convergence, Sungkyunkwan University, 2066 Seobu-ro, Jangan-gu, Suwon 16419, Korea; bg092@g.skku.edu
2. Department of Software, Sungkyunkwan University, 2066 Seobu-ro, Jangan-gu, Suwon 16419, Korea; rkdtks419@skku.edu
* Correspondence: jpjeong@skku.edu; Tel.: +82-10-9700-6284 or +82-31-299-4267

**Abstract:** Wearing a hard hat can effectively improve the safety of workers on a construction site. However, workers often take off their helmets because they have a weak sense of safety and are uncomfortable, and this action poses a large danger. Workers not wearing hard hats are more likely to be injured in accidents such as human falls and vertical falls. Therefore, the detection of wearing a helmet is an important step in the safety management of a construction site, and it is urgent to detect helmets quickly and accurately. However, the existing manual monitor is labor intensive, and it is difficult to popularize the method of mounting the sensor on the helmet. Thus, in this paper, we propose an AI method to detect the wearing of a helmet with satisfactory accuracy with a high detection rate. Our method selects based on YOLOv4 and adds an image super resolution (ISR) module at the end of the input. Afterward, the image resolution is increased, and the noise in the image is removed. Then, dense blocks are used to replace residual blocks in the backbone network using the CSPDarknet53 framework to reduce unnecessary computation and reduce the number of network structure parameters. The neck then uses a combination of SPPnet and PANnet to take full advantage of the small target's capabilities in the image. We add foreground and background balance loss functions to the YOLOv4 loss function part to solve the image background and foreground imbalance problem. Experiments performed using self-constructed datasets show that the proposed method has more efficacy than the currently available small target detection methods. Finally, our model achieves an average precision of 93.3%, a 7.8% increase over the original algorithm, and it takes only 3.0 ms to detect an image at 416 × 416.

**Keywords:** small-object detection; helmet detection; image super resolution; YOLO; YOLOv4; safety





## 1. Introduction

Construction site safety is more important than ever as more and more infrastructure needs to be built as the industry revitalizes. Accidents can be prevented by using personal protective equipment [1]. Helmets are the most important personal protective equipment to protect workers from falling objects [2], and it is legally mandatory at construction sites around the world to wear them [3]. However, wearing a hard hat tends to be neglected due to discomfort and a weak sense of safety. Therefore, it is very important to check whether the worker is wearing a helmet for the safety of the worker, and it is possible to increase the level of safety management. In existing helmet-wearing inspections at construction sites, surveillance image inspection and manned patrols are performed [4]. However, this method requires a lot of time and effort, and a manual monitor may cause misjudgment due to fatigue because the examiner must stare at the screen for a long time. Accordingly, image analysis techniques are rapidly developing with the help of new technologies and sensors to detect helmets at construction sites.

Although the number of deaths from industrial accidents has decreased compared to the past, the death rate in the construction industry is still high, and more than half

of all deaths occur in the construction industry. Not wearing a helmet at a construction site can lead to a fatal accident. Therefore, to prevent such fatal accidents, a system that recognizes and detects whether a helmet is worn at a construction site is required. Advances in computer technology have made it possible to train large-scale deep neural networks by applying GPUs for massively parallel computing [5,6].

Object detection is an essential capacity of computer vision solutions. It has gained attention over the last few years by using the core components of parallel-Self-Organizing Map (SOM) that are used for the classification of meteorological radar images [7]. Miller et al. used the congealing process to minimize the summed component-wise (pixel-wise) entropies over a continuous set of transforms on the images' data to demonstrate a procedure for effectively bringing test data into correspondence with the data-defined model produced [8]. In the field of object detection, a series of deep learning-based methods have been developed, and CNNs (Convolution Neural Networks) are the most used because of their excellent characteristics in high-level feature extraction. As a result, they are gradually replacing conventional detection methods in image analysis [9]. There are two main methods for CNN-based object detection. The first is a two-stage detector that first extracts a set of candidate regions where the object may be and then applies the CNN detector to object classification and location. Representatives include R-CNN (Region-Based Convolution Neural Network) [10] and improved networks such as Fast R-CNN [11] and Faster R-CNN [12]. On the other hand, the single-stage detector treats object detection as a regression problem, directly predicting class probabilities and bounding box coordinates according to CNN features. Representative networks are the single-shot multibox detector (SSD) [13], look-at-once (YOLO) [14], and the improved network [15]. The development of CNN-based detectors has motivated deep learning-based hard hat-wearing detection methods [16,17], and many researchers consider deep learning-based methods as essential measures to solve construction safety management problems [18].

We developed a helmetless autodetector based on Faster R-CNN, achieving an accuracy of 90.1% to 98.4% in various scenarios [19]. However, Faster R-CNN cannot meet the real-time requirement as it takes about 0.2 s to detect the image [20]. To improve Faster R-CNN, we used multi-scale training, augmented anchor strategy, and online hard-example mining. The safety helmet detection accuracy was finally improved by 7% compared to the existing algorithm, but the operating speed did not improve. Recently, many researchers have been working on single-stage detectors for hard hat detection tasks. Shi et al. [21] extracted multi-scale feature maps using the image pyramid structure and combined them with YOLOv3. In the research of Wu et al. [22], instead of the original backbone of YOLOv3, a densely connected convolutional network [23] was adopted, resulting in better detection results with the same detection time. Shen et al. [24] obtained a face-to-helmet regression model after detecting hard hats based on the first stage face detector [25]. Li et al. [2] chose the SSD algorithm to meet the real-time requirement and added MobileNet [26] to reduce the computational load. Wang et al. [4] proposed a new objective function to improve YOLOv3 and applied it to helmet detection. Single-stage detectors generally have lower two-stage detector accuracy but provide higher throughput [27]. YOLOv4 is suitable for real-time object detection by complying with both speed and accuracy among object detection models [28]. All the above studies show that developing a deep learning-based hard hat-wearing detection method can help reduce human and material resources, prevent omissions and false positives caused by human factors, and lay the foundation for the next step.

This paper proposes the image super revolution improved network based on YOLOv4 to solve the small helmet detection problem. The paper makes the following specific contributions:

- To solve the small detection problem, we proposed, based on YOLOv4, the object detection accuracy performance by increasing the resolution of low-resolution photos through the ISR module and the high performance in our model.

- To improve feature extraction, we proposed the CSP1-N network as the backbone feature extraction network to improve feature extraction.
- To make detailed feature fusion processes during training we propose the CSP2-N network in the neck.
- To train non-linear features, we propose the Hard Swish activation to improve the model.

The structure of this paper is organized as follows. Section 2 introduces the related works of safety helmet detection. Section 3 introduces the algorithm of the proposed model, and Section 4 presents the experimental environment, training details, and analysis of the results. In Section 5, we provide our conclusions.

## 2. Related Works

In this section, we will cover the general history and development of the YOLO algorithm and how it has evolved in terms of speed and accuracy. Further, we will briefly cover previous efforts to detect safety helmets on construction sites.

### 2.1. YOLO

The two main problems of object detection are classification and localization. When those two problems are solved separately, we call it a two-stage detector. RCNN is a typical two-stage algorithm. Two-stage algorithms have advantages in terms of accuracy, but it has slow processing speed and structural complexity. On the other hand, a one-stage detector solves the two before-mentioned problems in one process. YOLO is one such process. The one-stage detector YOLO is an algorithm that can be used as an end-to-end training method. It approaches the problems of categorizing and locating as a single regression problem. YOLO divides the input image into an S × S grid [14]. If the center of an object is within a specific grid, the grid takes on the role of detecting the object. In each grid, B bounding boxes are predicted. Confidence scores predicted for each B-Box have a value between 0 and 1, indicating the probability that an object exists in the corresponding bounding box. In addition, each grid also predicts the probability of classes. The final predicted value has a tensor value of size (S × S × (B*5 + C). Finally, the final bounding box is selected through non-max suppression. This method of YOLO shows a fast processing speed, but it is difficult to detect small objects or close objects because only two boxes and one category can be predicted in a region.

For the model to predict more boxes and show better performance, YOLOv2 borrows the concept of anchor boxes from Faster-RCNN [10]. When using an anchor box, you only need to predict the difference between the anchor box and the object. This makes it easier for YOLOv2 to solve problems, which leads to improved performance. In addition, YOLOv2 changed the resolution of the input image to 416 × 416 pixels so that the last feature map extracted was 13 × 13, which is an odd number so that an anchor box was created in the center. In this way, the object in the center can also be predicted well. The grid of YOLOv2 provides five anchor boxes, so the size standard of the anchor box is determined based on the anchor boxes clustered in the training set.

However, the YOLO family of algorithms shows weakness in small object detection. Therefore, YOLOv3 tried to solve this problem by using three different scales [29]. The three scales extract features at the pyramid level by adding a convolutional layer in darknet-53. The added convolutional layer is used to create a feature pyramid. Upsampling is performed twice on the final feature map, and the final feature map and the previous two feature maps are combined with the upsampled feature map. After that, the merged feature map is input to the Fully Convolutional Networks (FCN), and the same process is applied to the feature map in the next level.

YOLOv4 designed the model using the latest deep learning techniques. When trained based on a well-structured structure, it boasts very good performance in terms of speed and accuracy. Compared with YOLOv3, the biggest difference is some additional techniques in detecting and adjusting the network structure. Through this method, the performance of

AP is increased by 10% and FPS by 12% compared to the previous model [4]. In YOLOv4, the accuracy is increased by adding Bag of Freebies (BoF) and Bag of Special (BoS) techniques. BoF is a factor involved in learning, such as data augmentation, loss functions, and regularization, and refers to methods to increase accuracy by increasing the training cost. BoS mainly refers to techniques from an architecture point of view, including post processing and techniques to increase accuracy by only increasing the inference cost. YOLOv4 achieves an increase in processing speed by utilizing CSPDarknet in darknet-53, which was used in the previous version. CSPDarknet proposed a cross-stage partial network structure that can alleviate the very heavy inference costs and minimize the loss of accuracy. After dividing the input feature map into two parts, one part is merged from the back without participating in the operation. Based on this, inference cost, memory cost, etc., could be reduced. From the learning point of view, it is argued that the loss of accuracy is small because it has a good effect on learning by dividing up the gradient flow.

Figure 1 shows the network structure of YOLOv4. It consists of a backbone with CSPDarknet 53, a neck with a Path Augmented Network, and heads. The backbone extracts the feature map from the input image. The neck is a recently used structure placed between the backbone and the head. It is used to collect features from the various stages within the neck; it is built using a few bottom-up, top-down paths. The head predicts the category and bounding box of the object.

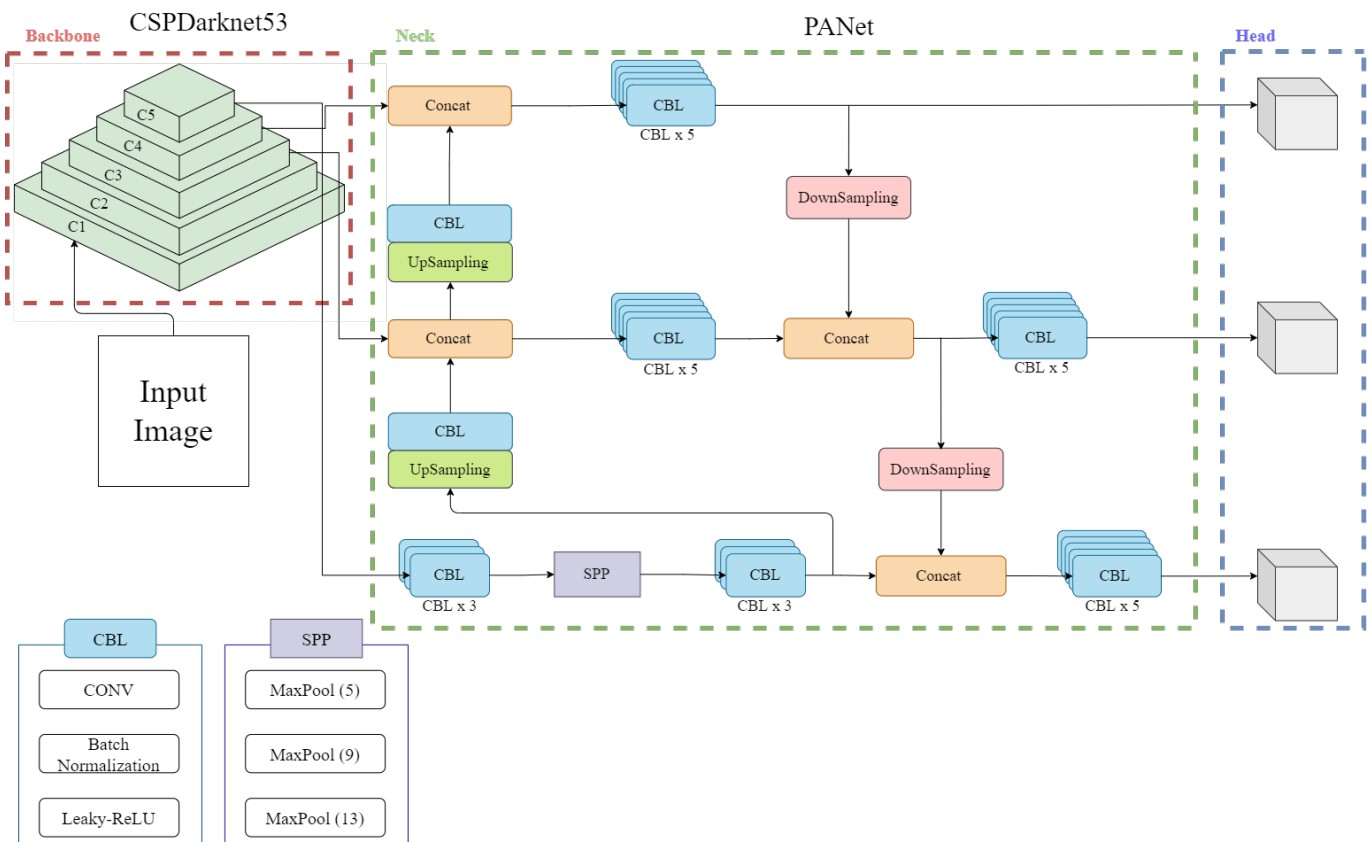

**Figure 1.** Network structure of YOLOv4.

### 2.2. Traditional Safety Helmet Detection

Accidents in constructions site can be significant when it damages human life. Especially when the brain is damaged, it can affect the construction worker permanently. Due to the crucial importance of safety at construction sites, detecting safety helmets has been worked on for past years. The detection of safety helmets mainly tried to detect outstanding features of the helmet, such as its color and shape.

Wen et al. [30] tried to detect a person wearing a safety helmet through the surveillance camera of an ATM. They applied a circle/circular arc detection method based on the modified Hough Transform. They placed an object that was taken by the surveillance camera in the circular arc/circle, and used geometric features to detect whether the safety helmet was in the arc.

Rubaiyat et al. [31] used the Histogram of Oriented Gradient (HOG) approach, a well-known human detecting algorithm, to detect construction workers. Further, they used Circle Hough Transform (CHT) with color to extract features of the safety helmets that workers put on. Hu et al. [32] detected human faces through skin color by using the YCbCr color model. By detecting human faces, they tried to locate safety helmets. To achieve this goal, they used wavelet transforms to preprocess the image, and extract features of safety helmets.

Liu et al. [33] located face regions by skin color detection, and by obtaining face regions, they were able to acquire the region image above the face. Further, they used the Hu moment to extract features and used neural networks and SVM to finish classification. Du et al. [34] purposed a safety helmet algorithm through a Deformable Part Model. With this model, they used Latent Support Vector Machine (SVM) to correctly train the algorithm to classify and detect safety helmets. YOLOv4 can be used in different fields in our society; Yu J. and Zhang W. use a face mask-wearing detection algorithm based on an improved YOLOv4 by rebuilding the YOLOv4 network [35]. Li et al. [36] detected moving objects in the power substation via the Vibe background modeling algorithm. Through the motion–object segmentation they acquired through the algorithm, they processed it into a C4, a real-time human classification framework. C4 allowed them to detect the workers with high accuracy and processing speed. They used data to locate the worker's head and used color transformation and the color feature.

## 3. Realtime ISR-YOLOv4 Based Small Object Detection

Yu and Zhang and Wang et al. used the small object detection algorithm proposed in this paper divided into input, backbone, network, neck network, and head. After image super resolution processing is performed on the input part to extract small objects, the backbone part is used to extract features of small objects from the image, the neck part is used for multi-scale features, and the head part is used for multi-scale features It uses a map to detect, then targets it and determine its location [35,37]. The major structure of small object detection is divided into four categories, so we used this structure in safety helmet detection by improving the backbone, neck, and prediction parts. The structure of the algorithm is shown in Figure 2.

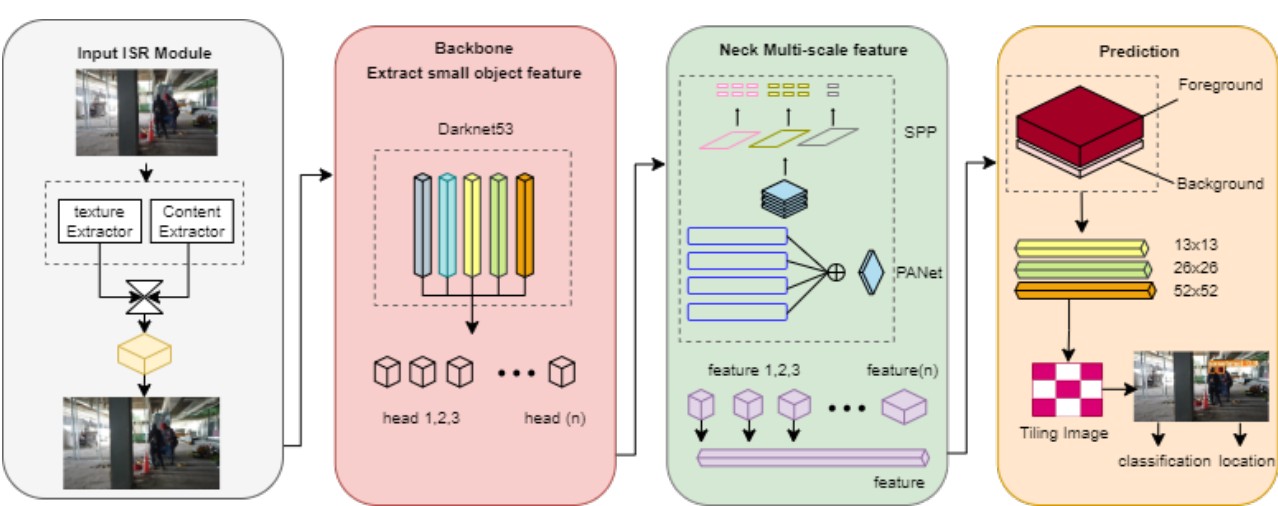

**Figure 2.** Proposed Helmet Object Detection Structure.

### 3.1. Image Super Resolution (ISR)

The proposed small object detection algorithm proceeds with input, backbone, neck, and head output. The input part performs ISR processing, extracts small object features from the backbone image, the neck fuses multi-scale features, and the prediction specifies detection using a multi-scale feature map. The algorithm is shown in Figure 2. The ISR module was added to the input to capture local details of small targets. The main extracted content used texture extraction for image enhancement and texture extraction for image identification. In the backbone network, blocks in Darknet53 are connected in the same way as each layer in DenseNet [23]. We use this specific association for training deeper network structures and for neuronal modes of functional maps learned at different levels. This connection can avoid overfitting with fewer parameters than other networks; the neck requires fewer parameters than other networks and avoids overfitting. The neck part maintains the PAnet structure and the original spatial pyramid pooling structure. PANet is a functional fusion module of this part, combining different scale functions. Spatial pyramid modules are structures added to the neck to amplify the usable fields of the network. YOLOv4 [28] was selected as the head, and a loss function was added to the foreground and background balance loss, reliability loss, and classification loss due to bounding box regression to improve the accuracy of small object detection.

The ISR module input is split into two parts, content and local textures, as shown in Figure 3. It is first extracted with a content extractor, and then sub-pixel convolution is used to double the resolution of the content feature. The texture extractor connects the two parts to the output terminals while selecting a trusted local texture from the base and reference and function and denoising the reference function [37]. P0 represents the output of the image super revolution module and is defined as:

$$P_0 = R_l(I_0 \otimes R_c(I_1) \uparrow_{2\times}) + R_c(I_1) \uparrow_{2\times} \tag{1}$$

$I_0$ is the local texture input, $I_1$ is the content input, $R_1()$ is the texture extraction factor, $R_c()$ is the content extraction factor, $\uparrow_{2\times}$ indicates the secondary up-scaling via sub-pixel convolution, and $(X)$ indicates feature stitching. Both the content extractor and the texture extractor consist of residual blocks. The default method uses sub-pixel convolution to perform advanced spatial resolution processing of the content features of the underlying input.

To increase the pixel values of width and height, subpixel convolution transfers the pixels in the channel dimension [37]. The features generated by the convolutional layer are expressed as:

$$F \in \mathbb{R}^{H \times W \times C \cdot r^2} \tag{2}$$

The pixel shuffling operation of subpixel convolution rearranges the features to rH × rW × C [37]. This operation is mathematically defined as:

$$PS(F)_{x,y,z} = F_{\lfloor x/\lambda \rfloor, \lfloor y/\lambda \rfloor, C \cdot \lambda \cdot \bmod (y,\lambda) + C \cdot \bmod (x,\lambda) + z} \tag{3}$$

The pixel-representing part of the output feature is $PS(F)x; y; z$, and the coordinates $(x; y; z), r$ after the pixel shuffling operation $PS()$, are the upscaling factors. ISR is where $PS(F)x; y; z$ represents the output feature pixel. The coordinates $(x; y; z)$ of the pixel shuffling operation $PS()$ are upscaling factors. The size of the space is doubled by using $D2$ in the ISR module. $D$ and content input $A$ are sent from the texture area to the texture extraction, which makes the extraction of small objects highly reliable. Adding textures and content on an element-by-element basis allows the output to incorporate semantic and local information from inputs and references. Thus, $P()$ has a similar meaning to the trusted texture selected from shallow feature $D$ and deeper level 11.

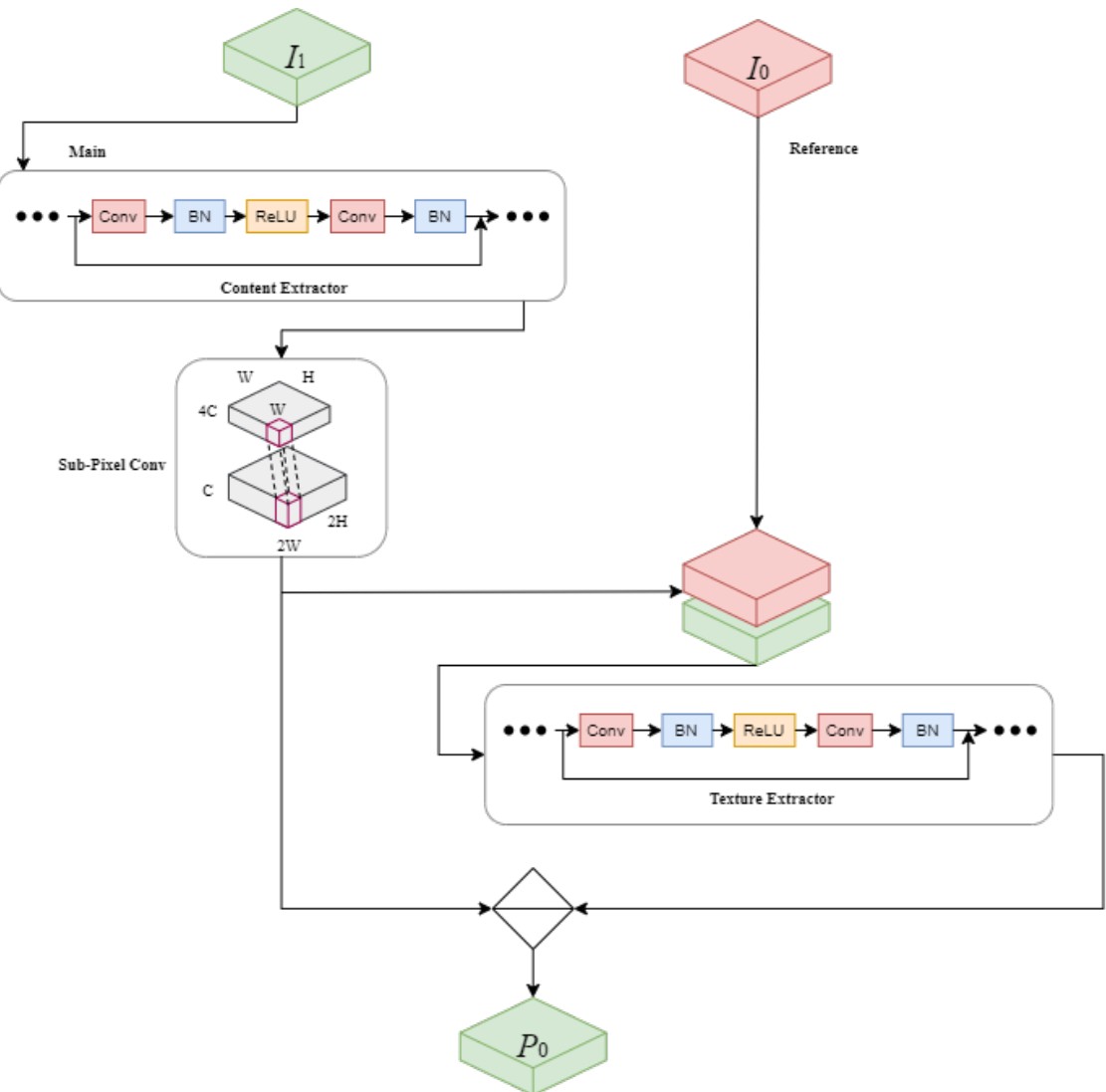

**Figure 3.** ISR-Module Framework.

*3.2. Backbone Network*

We add the remaining modules to YOLOv4 to reduce the parameters and improve the network learning ability. Yu and Zjang used CSPNDarkNet53 as a face mask-wearing detection module. The rest of the unit can be expressed as follows. First, there was a $1 \times 1$ convolution; we then proceeded with a $3 \times 3$ convolution, after which they added weights to both outputs of the module. The weights retain dimensional information, and the goal is to augment the information in the feature layers [35]. We used this method in helmet detection differently; by maintaining the first and last CSP connections of each extra residual network, inter-edges are added between every two adjacent extra blocks to provide cross-layer flow separation of gradients and accelerate forward propagation while simultaneously repeating deeply repeating extra blocks. It whitens wasted and vanishing resources that occur in between. After the image feature layer set CSPDarkNet53 is the input, it continues to perform convolutional downsampling to get better information. Therefore, the three layers at the end of the backbone have the best semantic information and configuration, and the last three layers are chosen as input to the SPPNet. The network structure of CSPDarkNet53 is shown in Figure 4.

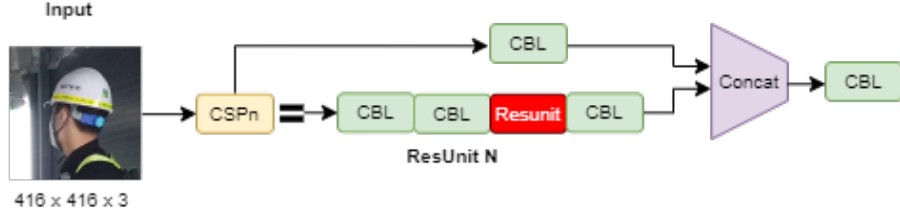

**Figure 4.** CSPDarknet Network Structure.

In this paper, CSPDarknet53 of YOLOv4 was changed to a CSP1-N module for increased performance. YOLOv4 uses redundancy networks to lower the computing performance requirements for the algorithm, but the memory requirements are partially improved with the CSP1-N module.

Compared to CSPDarkNet53 in Figure 4, it is an upstream network using an H-Swish function [38], as shown in the following equation

$$\text{h-swish}\,(x) = x\frac{\text{ReLU}\,6(x+3)}{6} \tag{4}$$

Since the Swish function [39] contains a sigmoid function compared to the ReLU function, the Swish function has higher computational performance requirements but better accuracy. In addition, the model runtime reduction slope error attenuation can be reduced through the H-Swish function. It has been reduced in past work [40]. It also improves the model object detection accuracy performance by segmenting the input layer of the image block in CSP1-N. As shown in Figure 5, it is used as the residual edge of the convolution operation.

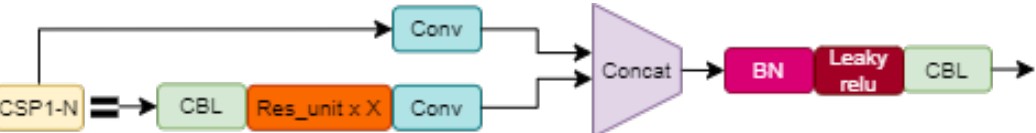

**Figure 5.** CSP1-N Network Structure.

*3.3. Neck Network*

Convolutional neural networks require the input images to be the same size. In conventional convolutional neural networks, fixed input values are obtained through truncation and warping operations. There is also a study in which Yolov4 uses multi-scale local function to improve the demand for fixed input size through SPPNet [41]. Yu and Zjang, by adding the CSP2-N module to the PANnet structure to combine full functional information and multiple scales, improved the model performance accuracy and functionality. CSP2-N is shown in Figure 6. The neck network of YOLOv4 adopts common convolution operation, and CSPNet has advantages such as excellent learning ability, computing bottleneck, and memory cost reduction. By improving the CSPNet network module based on YOLOv4, the network function convergence function can be further strengthened Ref. [35]. However, in our experiment, we add the CBL-Resunit structure for the information delivery networks to optimize the connectivity of the neck, and the main role is to use the cross-layer connectivity properties of ResNet to allow the information processing to be distributed through multiple paths in the FPN and PAN. Information and localization are effectively fused through different pathways to improve image processing. The SPPNet network composes functional fusion between other backbone layers at the neck by combining the bottom-to-top deep positioning function, and in PANet, the top-to-bottom calculation method for detailed functions is implemented through this convergence operation. It provides more useful features for predictive networks. The CSP2-N network is shown in Figure 6.

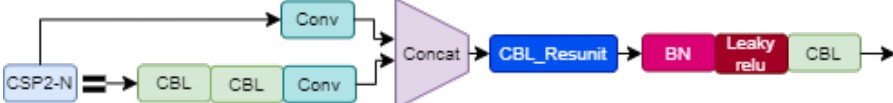

**Figure 6.** CSP2-N Network Structure.

### 3.4. Tiling Images

Tiling effectively magnifies the detector on small objects but can maintain the small input resolution needed to run fast inference. If you use tiling during training, it is important to remember that you need to tile the image at inference time for more accurate results. This is because we want to keep the zoomed perspective so that the object during inference is sized similar to the object during training. Here is a model trained to detect helmets via construction site photos. In Figure 7, the model was trained with tiling to better recognize helmets given the small size and large size of the source image, but if tiling was not used in the inference, instead of helmets, it will detect fittings and other large shapes. The object we tried to detect during training.

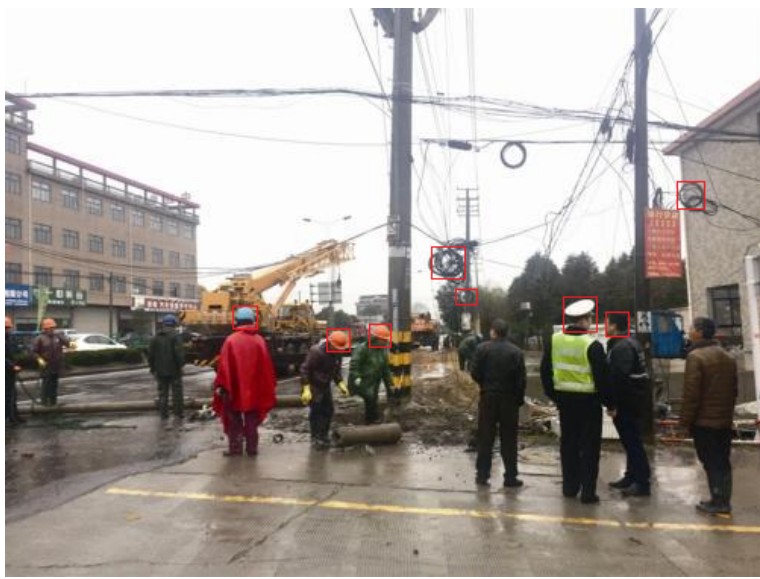

**Figure 7.** Tiled image during inference training.

Therefore, we tiled the image before running the inference. Figure 8 allows you to magnify parts of the image and make the helmet easier to detect against the model.

### 3.5. ISR-YOLOv4 Network Structure

The improved network model uses three CSP1-N networks in the singularity extraction network from the backbone, as shown in Figure 9, and each CSP1-N network has N remaining units. In this paper, to reduce the computational requirements, the residual modules are connected in series with N residual unit combinations. This method can modify two 3 by 3 convolution operations with 1 by 1, 3 by 3, 1 by 1 convolution modules. The first 1 by 1 convolutional layer can reduce parameters while reducing the number of channels to approximately 50%. A $3 \times 3$ convolutional layer can improve feature extraction and reuse the residual number of channels. Finally, the $1 \times 1$ convolution operation recovers the output of the $3 \times 3$ convolution layer, so the alternative convolution operation is efficient and has high accuracy for feature extraction and can reduce the computer performance requirements.

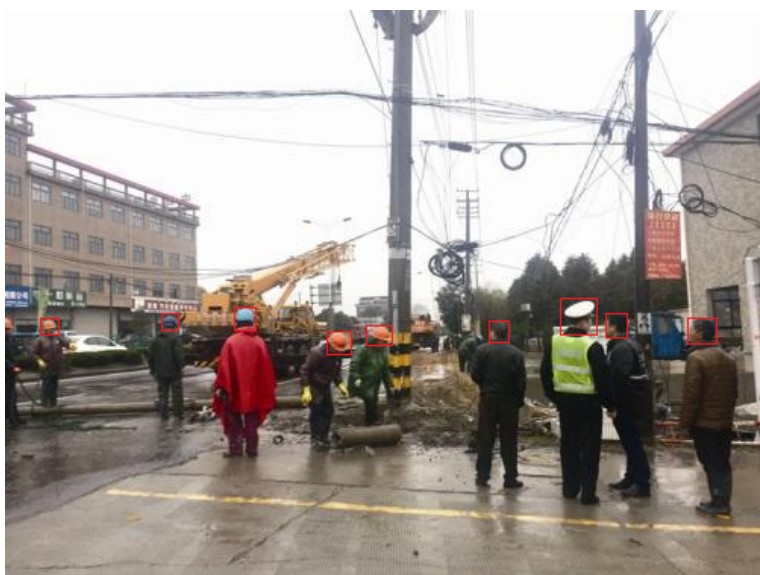

**Figure 8.** Tiled image before inference training.

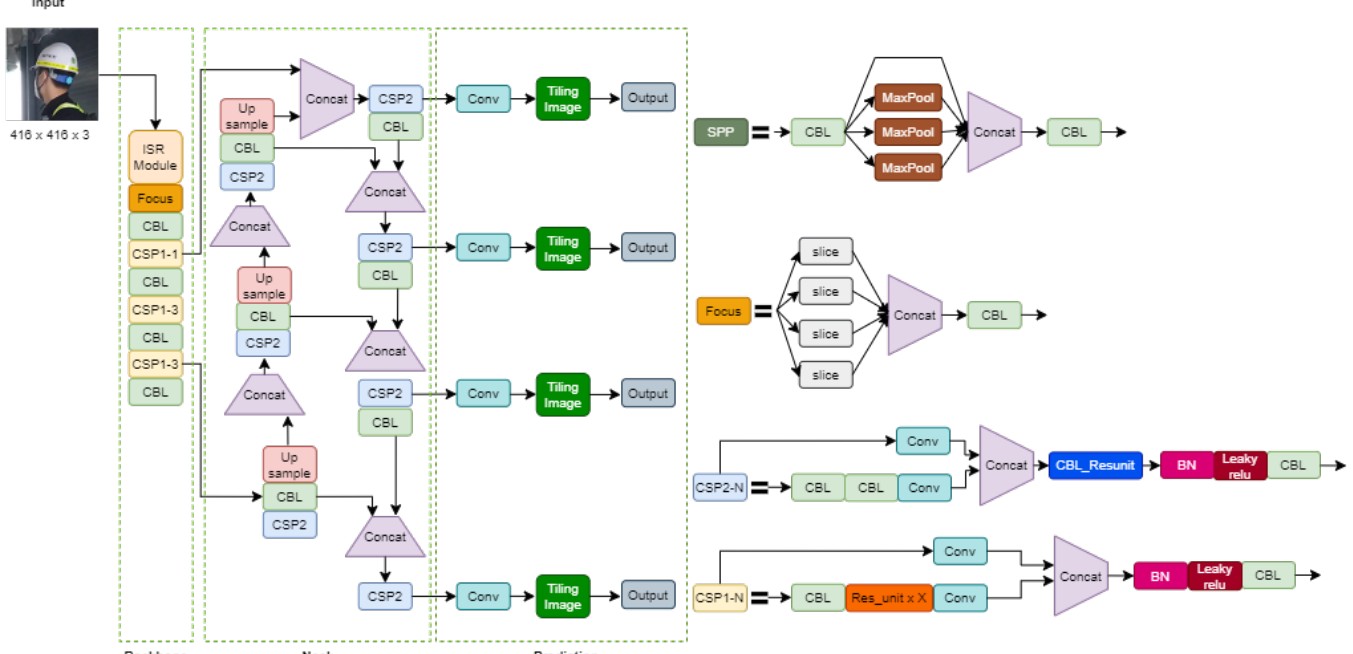

**Figure 9.** Improved YOLOv4 Network.

## 4. Experiment and Results

### 4.1. Experiment Environment

All experiments and evaluations in this study were conducted on a 52 GB RAM and NVIDIA Tesla V100 16 GB GPU. An anomaly detection model was built using PyTorch version 1.10.0 + cu111. The learning hyperparameters were 300 epochs, batch size 8, learning rate $10^{-5}$, and patch size 64. Table 1 shows the H/W and S/W Specification.

**Table 1.** H/W and S/W Specification.

| H/W and S/W | Specification |
|---|---|
| System | Window 11 |
| CPU | Intel Core i9 11900K |
| Memory | 52 GB |
| GPU | Nvidia Tesla V100 16 GB |
| OS Type | 64-bit |
| CUDA | version 11.0 |
| Pytorch | version 1.5.1 |

We also considered the problem of object detection to improve model performance by objectively collecting helmet and head images with various scenarios from the web and mostly on construction sites. Our dataset is rich as it learns by considering the manufacturing environment and outdoor and indoor factors sufficiently. Moreover, we collected various small helmet and head images for increased model precision. Therefore, the model algorithm has excellent helmet detection ability not only in the manufacturing environment but also in various environments. In this experiment, the head of each figure is each label. Label 0 is "head" without a hard hat. Label 1 indicates a "helmet", a head with a hard hat, namely two categories of detection. Table 2 shows the datasets used in the experiments.

**Table 2.** Dataset.

| Category | Train Set | Test Set | Validation Set |
|---|---|---|---|
| helmet | 7185 | 1532 | 512 |
| head | 7025 | 1512 | 645 |

*4.2. Performance Metric*

The precision (P) and recall (R) curves were referenced as a function of precision (*y*-axis) and recall (*x*-axis) for different probability thresholds. Comparisons with single variables and comparisons of different models allowed a more holistic understanding of the relationship between precision and recall. Precision and recall are expressed by Equation (5).

$$\begin{aligned} \text{precision} &= \frac{TP}{TP + FP} \\ \text{recall} &= \frac{TP}{TP + FN} \end{aligned}$$
(5)

$TP()$ is defined as the number of individuals that correctly predicted true positives for each class, and $FP()$ is defined as the number of individuals whose content was predicted differently. False negatives ($FN$) represent the number of correct answers that the model did not predict. Object detection studies generally show that the closer the curve is to the upper right corner, the better the model's performance. *Mean Average Precision*, which is the average value of the *Average Precision* of the entire class, is expressed by Equation (6).

$$Mean\ Average\ Precision = \frac{\sum_{i=1}^{n} Average\ Precision_i}{n}$$
(6)

In Equation (6), $n$ is the number of classes, and the average precision of each class corresponds to the area under the precision-recall curve. In general, the higher the mean average precision, the better the model's performance. However, there is a trade-off between precision and recall in Equations (5) and (6). The detection capability evaluation metrics *F*-Mesure is expressed as follows.

$$F_\alpha = \frac{(\alpha^2 + 1) \times P \times R}{\alpha^2 (P + R)}$$
(7)

The precision ratio and recall mean are expressed as $F_1$ in Equation 8 when

$$F_1 = \frac{2 \times P \times R}{(P + R)} \tag{8}$$

### 4.3. Results

### 4.3.1. Hyperparameters of the Model

All experiments were performed under the following parameters epoch 300, batch size 8, and input image size 416 × 416 × 3. The parameter tuning process is shown in Table 3.

**Table 3.** Hyperparameters of the Model.

| Hyperparameters | Value |
| --- | --- |
| width | 416 |
| height | 416 |
| channels | 3 |
| momentum | 0.949 |
| decay | 0.0005 |
| angle | 0 |
| saturation | 1.5 |
| exposure | 1.5 |
| hue | 0.1 |
| learning rate | 0.00261 |
| max batches | 500,500 |

### 4.3.2. Model Size and Training Time

Model sizes are 0.371× and 0.387× for YOLOv4 and YOLOv3. Under the same conditions, the training time of this model was 2.945 h, which was the lowest among the YOLO and SSD models in the experiment, followed by YOLOv6. Faster R-CNN requires a lot of computing power because the model is applied by generating W × H × K candidate regions using RPN (Region Proposal Network). Meanwhile, Faster R-CNN uses the full connection layer in the ROI pooling layer while keeping the ROI pooling layer the same, and many iterations occur in the network, reducing the learning speed of the model, as shown in Table 4.

**Table 4.** Comparison of different models in parameters, model size, and training time.

| Models | Parameters | Model Size | Training Time |
| --- | --- | --- | --- |
| SSD | 25.4 MB | 92 MB | 3.452 h |
| YOLOv3 | 59.9 MB | 198 MB | 7.951 h |
| YOLOv4 | 62.5 MB | 201 MB | 9.267 h |
| YOLOv-v5 | 65.3 MB | 213 MB | 10.152 h |
| YOLOv-v6 | 47.3 MB | 132 MB | 3.012 h |
| Proposed Work | 46.8 MB | 120 MB | 2.945 h |

### 4.3.3. Inference Time

In this paper, the real-time performance of the test was verified. Since FPS (frames per second) is often used to measure the real-time performance of a model, it can be seen from Table 5 that the higher the FPS, the better the real-time performance.

**Table 5.** Comparison of inference time and real-time performance.

| Models | Inference Time |
|--------|----------------|
| SSD | 96.0 s |
| YOLOv3 | 156.8 s |
| YOLOv4 | 158.5 s |
| YOLOv5 | 160.2 s |
| YOLOv6 | 147.8 s |
| Proposed | 148.9 s |

### 4.3.4. Mean Average Precision

As shown in Table 6, our model has a Mean Average Precision (mAP) 0.15 higher than Faster R-CNN mode, 0.118 higher than SSD, 0.107 higher than YOLOv3, 0.078 higher than YOLOv4, 0.039 higher than YOLOv5, and 0.003 higher mAP than YOLOv6.

**Table 6.** mAP comparisons of different models.

| Model | Helmet AP | Head AP | mAP |
|-------|-----------|---------|-----|
| SSD | 0.819 | 0.811 | 0.815 |
| YOLOv3 | 0.824 | 0.828 | 0.826 |
| YOLOv4 | 0.858 | 0.853 | 0.855 |
| YOLOv5 | 0.891 | 0.898 | 0.894 |
| YOLOv6 | 0.928 | 0.932 | 0.930 |
| Proposed | 0.932 | 0.935 | 0.933 |

Through this experiment, we fully demonstrate that our model outperforms YOLOv4 and SSD in comprehensive performance and slightly outperforms the recently released YOLOv6. However, the inference time was longer than that of YOLOv6. Faster R-CNN's mAP is higher than that of YOLOv4 and YOLOv3, but it has the lowest FPS, so the accuracy is high, but being difficult to use as real-time detection is a common characteristic of two-step detection algorithms.

As shown in Figure 10, our model detected safety helmets and the head well, both in real-time.

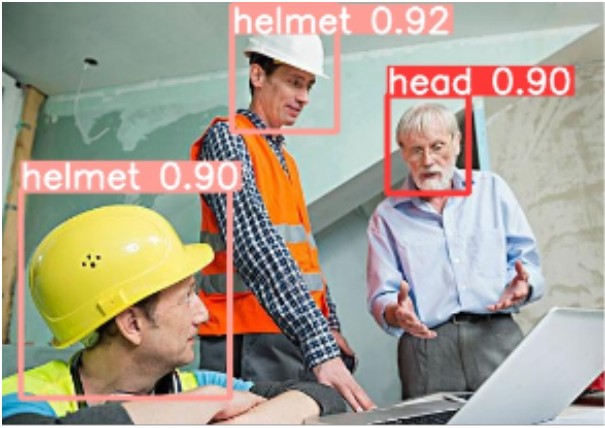

**Figure 10.** Sample test data on real-time video from our proposed model, with non-worn safety helmets and worn safety helmets.

As shown in Figure 11, our model detected non-wearing safety helmets even at a long distance well.

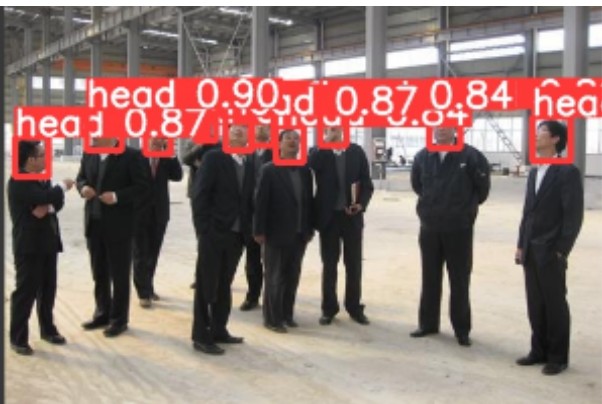

**Figure 11.** Sample test data on real-time video from our proposed model on small targets.

As shown in Figure 12, our model detected safety helmets at a long distance well.

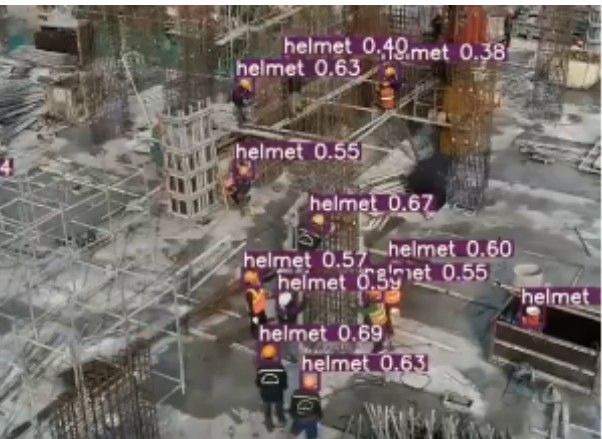

**Figure 12.** Sample test data on real-time video from our proposed model from a drone angle view.

As shown in Figure 13, our model detected safety helmets at a long distance well.

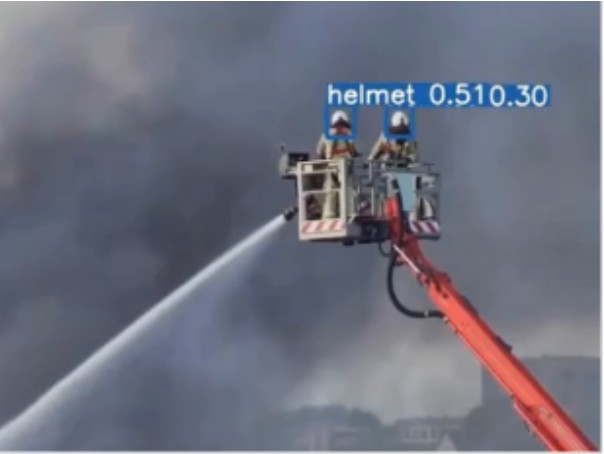

**Figure 13.** Sample test data on real-time video from our proposed model on small target firemen.

As shown in Figure 14, our model detected and classified the people who were wearing the safety helmets or not well, even if there are a lot of people.

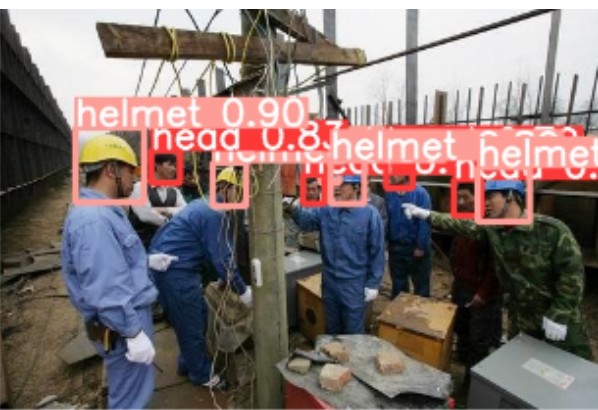

**Figure 14.** Sample testdata on real-time video from our proposed model who are wearing and not wearing helmets.

As shown in Figure 15, our model detected safety helmets from the above view in a busy construction site well.

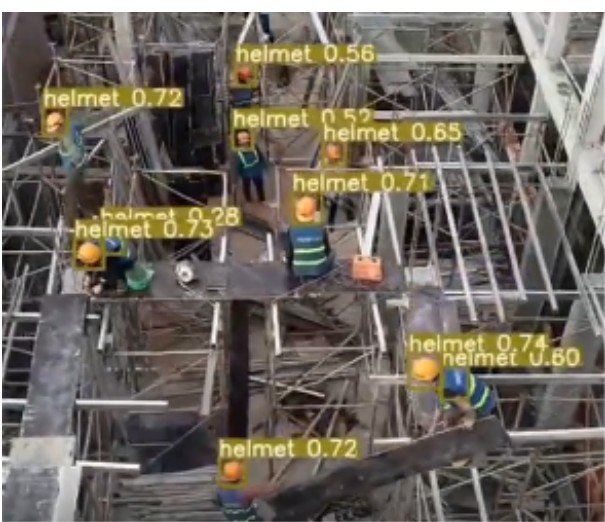

**Figure 15.** Sample test data on real-time video from our proposed model from the above angle.

As shown in Figure 16, our model detected non-worn safety helmets well in a domain different from construction sites.

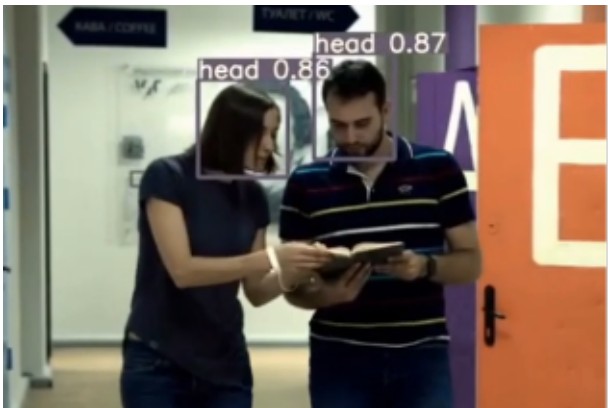

**Figure 16.** Sample test data on real-time video on our proposed model indoors.

## 5. Conclusions

In this experiment, we propose an improved network based on YOLOv4 to resolve the helmet detection problem. Meanwhile, the efficiency and robustness model were verified through comparative studies on object detection algorithms. First, we improved the object detection accuracy performance by increasing the resolution of low-resolution photos through the ISR module. The backbone feature extraction network was improved through CSP-1N module feature extraction, and CSP2-N was used in the neck part so that the model can handle parallel process learning. Further, to improve the model learning non-linear functions, the H-Swish activation function was added.

As a result of the experiment, our method in this paper showed the best performance detection accuracy on safety helmet detection compared to other algorithms. In addition, the algorithm also reduces the model's requirements for training cost and model complexity, allowing the model to be deployed to medium-sized devices and used in other industries where helmet-wearing decisions are required.

However, in this study, there is still a problem of insufficient feature extraction for difficult-to-detect samples or missing and false-positive cases, and it is difficult to collect a lot of data according to the Personal Information Protection Act. In addition, there is still a point where the type of helmet cannot be identified. Therefore, the next step should be extended to more object detection tasks by extending the dataset to helmet types and gaining further improvements to the model in the current work.

**Author Contributions:** Conceptualization, B.K. and J.J.; methodology, B.K.; software, B.K.; validation, B.K. and J.J.; formal analysis, B.K.; investigation, B.K.; resources, J.J.; data curation, K.K.; writing—original draft preparation, B.K. and K.K; writing—review and editing, J.J.; visualization, B.K.; supervision, J.J.; project administration, J.J.; funding acquisition, J.J. All authors have read and agreed to the published version of the manuscript.

**Funding:** This research was supported by the MSIT (Ministry of Science and ICT), Korea, under the ITRC (Information Technology Research Center) support program (IITP-2022-2018-0-01417) supervised by the IITP (Institute for Information & Communications Technology Planning & Evaluation). Also, this work was supported by the National Research Foundation of Korea (NRF) grant funded by the Korea government (MSIT) (No. 2021R1F1A1060054).

**Acknowledgments:** This research was supported by the SungKyunKwan University and the BK21 FOUR (Graduate School Innovation) funded by the Ministry of Education (MOE, Korea) and National Research Foundation of Korea (NRF) and the ITRC(Information Technology Research Center) support program(IITP-2022-2018-0-01417) supervised by the IITP(Institute for Information & communications Technology Planning & Evaluation).

**Conflicts of Interest:** The authors declare no conflict of interest.

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
