# Peer review of "Real-Time ISR-YOLOv4 Based Small Object Detection for Safe Shop Floor in Smart Factories"

_electronics, doi:10.3390/electronics11152348_

Round 1
Reviewer 1 Report
Dear authors,
1. Please update Table 6 of your paper by adding YOLO-v6 (https://github.com/meituan/YOLOv6)
2. Would you consider including a GitHub repository with your dataset and code?
Reviewer 2 Report
The manuscript reported a research related to Machine Learning methods to detect wearing a helmet with satisfactory accuracy with a high detection rate, especially by YOLO. The research was well conducted and the proposed system works well with good performance. The manuscript is suitable to publish in <<Electronics>>, but the actual version needs to be revised based on the following observations.
Object detection is an essential capacity of computer vision solutions. It has gained attention over the last years by using a core components of the “Once learning” and “Few-shot learning” mechanism for training by a few examples (Weigang and Silva, 1999; Miller et al., 2000; Li FeiFei et al., 2004). It is better for authors to describe these references
Line 69 with in-consistence: Li et al. [2]We chose …
Line 73-74 with English program problem: YOLOv4 that can use a sufficiently fast computer with satisfactory accuracy considering the balance between speed and accuracy [27].
Lines 82-89: the contributions are not well described: 1) It can not just mention what models you proposed, more important the performance of your proposed models. 2) the texts are not with one format: To solve small detection problem, We proposed the … We should be we
Line 103: two beforementioned
Line 125 English program problem: First, add a convolutional layer in darknet-53.
Line 209 there are no I0, I1 and Rt() in equation 1: I0 is local texture input, I1 is content input, Rt() is texture extraction factor,
There are too many typos in paper. The notations and presentation of the equations are very confused. It is better to be revised carefully.
References:
[1] Li Weigang, L. and N.C. da Silva, 1999, July. A study of parallel neural networks. In IJCNN'99. International Joint Conference on Neural Networks. Proceedings (Cat. No. 99CH36339) (Vol. 2, pp. 1113-1116). IEEE.
[2] Miller, E.G., Matsakis, N.E. and Viola, P.A., 2000, June. Learning from one example through shared densities on transforms. In Proceedings IEEE Conference on Computer Vision and Pattern Recognition. CVPR 2000 (Cat. No. PR00662) (Vol. 1, pp. 464-471). IEEE.
[3] Fei-Fei, Li, Rob Fergus, and Pietro Perona. "Learning generative visual models from few training examples: An incremental bayesian approach tested on 101 object categories." In 2004 conference on computer vision and pattern recognition workshop, pp. 178-178. IEEE, 2004.
Reviewer 3 Report
In this paper, the authors present a method for detecting helmets in images of workers at construction sites to improve worker safety. The proposed method uses an advanced YOLO-v4 algorithm with CSP1-N and CSP2-N modules and an H-swish activation function to increase the small object detection accuracy and reduce the computation time.
There is one major problem with this paper, and that is that the parts of the paper in which the authors describe their method are almost identical in content to the parts of the paper:
„Yu J., Zhang W. Face Mask Wearing Detection Algorithm Based on Improved YOLO-v4. Sensors (Basel). 2021 May 8;21(9):3263. doi: 10.3390/s21093263.“
The authors did not cite the above work or indicate how their method differs from that described in Yu and Zhang's paper.
Author Response
Thanks for the great observation from the reviewer. Our authors found that the reviewers’ the difference of our paper is on safety helmet detection not on the mask detection and they didn't use the ISR(Image Super Resolution) on their experiment and we use our private data.

Round 2
Reviewer 1 Report
Maybe you can not upload all the data. However, the code with some examples will be great for our scientific community.
Reviewer 2 Report
The revised version is improved very well. There are still some minor modifications. Please take care of the RED parts:
1) It has gained attention over the last years by using a core components of the few shot learning by parallel- SOM used for classification of meteorological radar images [40]. Miller et al. used congealing process to minimizing the summed component-wise (pixel- wise) entropies over a continuous set of transforms on the images data to demonstrate a procedure for effectively bringing test data into correspondence with the data-defined model produced [41].
2) Li et al. [2] chose the SSD algorithm to meet the real-time requirement
3) Shi et al. [20] extracted multi-scale feature maps using the image pyramid structure and combined them with YOLO v3. In the research of Wu et al. [21], instead of the original backbone of YOLO v3, a densely connected convolutional network [22] was adopted, resulting in better detection results with the same detection time. Shen et al, [23] obtained a face-to-helmet regression model
4) … computational load. Wang et al. [4] proposed a new objective function to improve YOLO v3 and applied it to helmet detection.
5) It is better to improve the quality of Figure 4, 5 and 6.
6) It is better to check the equation (4).
Reviewer 3 Report
Although in the revised version the authors cite the paper [A] in the chapter "Related Work", they do not point out in any way the great similarity of their solution with the solution presented in this work, i.e. they do not explain in the paper how their solution differs from the one presented in [A]. In fact, the authors claim that they have improved the performance accuracy and functionality of the model by adding the CSP2-N module to the PANnet structure (chapter 3.3., page 8, lines 279-281), but they have taken this solution from [A]. The text itself describing individual parts of the proposed helmet recognition system is still very similar to the text of Yu and Zhang's paper. The text in Section 3.2 of this paper is very similar to Section 4.1 of Yu and Zhang's paper, 3.3. is similar to 4.2, and 3.4 is similar to 4.4. The authors should modify this text and cite [A], where they explain parts of the algorithm that they have taken from [A].
Moreover, the proposed algorithm has also a great similarity with the solution presented in [B], and this paper is also not cited. There is a great similarity between the structure of the algorithms proposed in both papers (Figure 2 and Figure 3), and the text describing the solution itself is also very similar to the text in [B] (chapter 3.1). Due to the large number of publications that appear every day, there is always a possibility that scientists may overlook an important paper that needs to be cited. But in this case, the high similarity of the text describing the solution indicates that the authors knew about the [B], which makes the non-citation more problematic. According to the structure of the algorithm shown in Figures 1 and 2, there is no difference between the solution proposed in this paper and the one in the paper [B]. The authors should state that their algorithm is largely the same as the algorithm in [B], and explain how their algorithm differs and what they have improved by introducing these differences. They should also cite the paper [B] in Chapter 3 when describing the algorithm, as well as with respect to equations (1) , (2) and (3), which are taken from [B].
[A] Yu J, Zhang W. Face Mask Wearing Detection Algorithm Based on Improved YOLO-v4. Sensors (Basel). 2021 May 8;21(9):3263. doi: 10.3390/s21093263.
[B] Z. Wang, K. Xie, X.-Y. Zhang, H.-Q. Chen, C. Wen, and J.-B. He,‘‘Small-object detection based on YOLO and dense block via image super-resolution,’’IEEE Access, vol. 9, pp. 56416–56429, 2021.
In addition, it is necessary to correct the following:
· The sentence in line 10 “Afterward, the image resolution as well as the noise in the image are removed.” is incorrect because image resolution can not be removed
· Line 18 – instead of 416 416 there should be 416x416 pixels
· Lines 122-123 – Instead of “In addition, YOLOv2 changed the resolution of the input image to 416 so that the last feature map extracted was 13.. “ I suggest “In addition, YOLOv2 changed the resolution of the input image to 416x416 pixels, so that the last feature map extracted was 13x13.”
· Line 43 – The full name for the abbreviation SOM is missing.
· Line 134 - The full name for the abbreviation FCN
· Line 179 - The full name for the abbreviation SVM is missing.
· The sentence in lines 258-259 „This high computing power requirement can be met by using H-swish function through Howard’s mobile device, reducing the number of memory accesses per model and additional time and cost“ is a modified sentence from [A], but the sense of the original sentence is lost.
· The text in lines 265-269 is the same as the text in lines 270-274.
· Chapter 4 - The data set should be better explained.
· Chapter 4.2 - the abbreviations MAP, AP, P, and R should be listed next to the full name of the term when it is first mentioned in the text.
· In equation (7), the constant alpha appears, and in the text this constant is referred to as "a" (line 331 - "The precision ratio and recall mean are expressed as F1 expression 8 when a =1.").
· Chapter 4.3 should be rewritten in order that explanation of the results be clearer. Chapter 4.3 should be rewritten to make the explanation of the results clearer.
· The sentence in lines 348-349 is not clear, it should be rewritten “As shown in the table below, our model is 0.15 higher than the Faster R-CNN model, 0.118 higher than SSD, 0.107 higher than YOLOv3, 0.078 higher than YOLOv4, 0.039 higher
· The headings of Figures 8 through 15 should not be identical. Also, the text explaining these figures should be more informative.
· The text in lines 375-379 is the same as the text in lines 380-384.
There are many sentences in the paper that are either grammatically or linguistically incorrect and unclear. In addition to the problems mentioned above, this contributes to the low quality of this paper.
Round 3
Reviewer 3 Report
The authors have improved the text sufficiently, but there are still some deficiencies that they need to correct. To facilitate these corrections, I have suggested the changes as follows:
- Line 10 – Instead of „Afterward, the noise image as well as removed“ I suggest the following sentence „Afterward, the image resolution is increased and the noise in the image is removed“
- Line 43 – I suggest “…Self Organizing Map (SOM) …”
- Line 134 - I suggest “…Fully Convolutional Networks (FCN)…”
- Line 179 - I suggest “…Support Vector Machine (SVM)…”
- Line 265-267- I suggest deleting the sentence „These high computing power requirements allow Howard’s mobile devices to use H-swish capabilities to reduce the number of memory accesses per model and additional time and resources required.“ The statement in this sentece is incorrect and the sentence itself is not important to the rest of the text.
- Line 337 – I suggest „Precision (P) and recall (R) are expressed by Equation (5).“
Line 342 – In the sentence „The precision ratio and recall mean are expressed as F1 expression 8 when a =1.“ letter a should be changed to a
Lines 362-365 – IInstead of the current text, I propose the following: "As shown in Table 6, our model has a Mean Average Precision (mAP) 0.15 higher than Faster R-CNN mode, 0.118 higher than SSD, 0.107 higher than YOLOv3, 0.078 higher than YOLOv4, 0.039 higher than YOLOv5, and 0.003 higher mAP than YOLOv6."
